# Oral Ingestion of Synthetically Generated Recombinant Prion Is Sufficient to Cause Prion Disease in Wild-Type Mice

**DOI:** 10.3390/pathogens9080653

**Published:** 2020-08-13

**Authors:** Chenhua Pan, Junwei Yang, Xiangyi Zhang, Ying Chen, Shunxiong Wei, Guohua Yu, Yi-Hsuan Pan, Jiyan Ma, Chonggang Yuan

**Affiliations:** 1Key Laboratory of Brain Functional Genomics (Ministry of Education and Shanghai), Institute of Brain Functional Genomics, School of Life Sciences and the Collaborative Innovation Center for Brain Science, East China Normal University, Shanghai 200062, China; pchpanda@126.com (C.P.); 18918057893@163.com (J.Y.); kylezhangxy418@gmail.com (X.Z.); chenying19941228@163.com (Y.C.); w10861228@163.com (S.W.); yihsuanp@gmail.com (Y.-H.P.); 2Fujian Provincial Key Laboratory for the Prevention and Control of Animal Infectious Diseases and Biotechnology, School of Life Sciences, Longyan University, Longyan 364012, China; xiaoqsh2003@163.com; 3Center for Neurodegenerative Science, Van Andel Institute, Grand Rapids, MI 49503, USA

**Keywords:** prion, recPrP^Sc^, oral transmission, prion disease

## Abstract

Prion disease is a group of transmissible neurodegenerative disorders affecting humans and animals. The prion hypothesis postulates that PrP^Sc^, the pathogenic conformer of host-encoded prion protein (PrP), is the unconventional proteinaceous infectious agent called prion. Supporting this hypothesis, highly infectious prion has been generated in vitro with recombinant PrP plus defined non-protein cofactors and the synthetically generated prion (recPrP^Sc^) is capable of causing prion disease in wild-type mice through intracerebral (i.c.) or intraperitoneal (i.p.) inoculation. Given that many of the naturally occurring prion diseases are acquired through oral route, demonstrating the capability of recPrP^Sc^ to cause prion disease via oral transmission is important, but has never been proven. Here we showed for the first time that oral ingestion of recPrP^Sc^ is sufficient to cause prion disease in wild-type mice, which was supported by the development of fatal neurodegeneration in exposed mice, biochemical and histopathological analyses of diseased brains, and second round transmission. Our results demonstrate the oral transmissibility of recPrP^Sc^ and provide the missing evidence to support that the in vitro generated recPrP^Sc^ recapitulates all the important properties of naturally occurring prions.

## 1. Introduction

Prion diseases, also known as transmissible spongiform encephalopathies, are a group of fatal neurodegenerative diseases including Creutzfeldt-Jakob disease (CJD), fatal familial insomnia (FFI), and Kuru in humans, scrapie in sheep and goats, chronic wasting disease (CWD) in cervids, and bovine spongiform encephalopathy (BSE) in cattle [1,2]. Different from other late age onset neurodegenerative disorders, prion disease is a naturally occurring infectious disease that can be transmitted within and between species [2]. It is now well established that in prion disease, the host-encoded PrP converts from its normal, soluble and protease sensitive PrP^C^ form to the disease-associated, aggregated and protease resistant PrP^Sc^ conformer [3]. The prion hypothesis posits that PrP^Sc^ is the infectious agent and because of its seeding capability, PrP^Sc^ is able to seed the conversion of host-encoded PrP^C^ to PrP^Sc^, resulting in prion disease [1].

Using purified recombinant PrP (recPrP) plus defined non-protein cofactors, we previously showed that recPrP^Sc^ can be generated in vitro with serial protein misfolding cyclic amplification (sPMCA) technique [4,5]. Similar to naturally occurring prions, recPrP^Sc^ is aggregated, highly resistant to proteinase K (PK) digestion, and able to chronically infect susceptible cell lines. Importantly, recPrP^Sc^ has a high titer of infectivity [6,7], causing bona fide prion disease in wild-type mice through i.c. or i.p. inoculation [4,5,6]. The pathogenic process of recPrP^Sc^-caused disease is identical to those observed in naturally occurring prion disease [6]. Biophysical analyses revealed that the conformation of recPrP^Sc^ is similar to that of native PrP^Sc^ [8] and the unique recPrP^Sc^ conformation determines its pathogenicity in animals [9,10].

The studies of recPrP^Sc^ provide strong experimental evidence to support a causative role of PrP^Sc^ in prion disease. But whether the in vitro generated recPrP^Sc^ is able to cause prion disease in wild-type animal via oral route remains unclear. Oral ingestion is a well-established route for the transmission of many naturally occurring prion diseases, including the outbreaks of Kuru and variant CJD in humans, scrapie in sheep and goats, CWD in cervids and BSE in cattle [2]. Establishing the oral infectivity of recPrP^Sc^ would provide a critical evidence to unambiguously support that prion is the infectious agent for these naturally occurring prion diseases. In this study, we determined the oral transmissibility of recPrP^Sc^ and showed that a single oral ingestion of recPrP^Sc^ is sufficient to cause prion disease in wild-type mice.

## 2. Results

### 2.1. A Single Oral Feeding of RecPrP^Sc^ Causes Prion Disease in Wild-Type Mice

To determine whether recPrP^Sc^ is able to cause prion disease via oral route, we prepared recPrP^Sc^ with sPMCA. The sPMCA substrate without going through sPMCA reaction was used as a negative control. Brain homogenates prepared from a mouse that succumbed to prion disease (DBH, representing prion diseased brain homogenates) was used as a positive control. The presence or absence of PK-resistant PrP in these samples was verified by PK digestion and immunoblot analysis (Figure 1A).

For oral feeding, each mouse was individually housed in a clean cage without food and bedding. A mouse food pellet doused with recPrP^Sc^, sPMCA substrate or DBH was placed on the floor of the cage. After complete ingestion of the food pellet, the mouse was returned to its home cage and monitored for the development of signs of neurodegeneration. Intracerebral inoculation of the same batch of recPrP^Sc^ or DBH was performed as positive controls to demonstrate the prion infectivity in these preparations.

As expected, all mice that received i.c. inoculation of recPrP^Sc^ or DBH developed neurological signs of prion disease, including clasping, tail plasticity, hypokinesia, kyphosis and ataxia, and reached terminal stage at 196.2 ± 3.59 and 176.4 ± 5.17 days post inoculation (dpi), respectively (Figure 1B and Table 1).

Around 250 days after oral feeding, clasping was observed in one mouse (#9) from recPrP^Sc^-fed group and another mouse (#19) from DBH-fed group. Both mice developed unsteady gait, tail plasticity, kyphosis and weight loss, but urinary incontinence was only observed in #19 mouse. Disease progressed quickly and both mice reached terminal stage in about 40 days (Figure 1B and Table 1). Except for these two mice, no sign of prion disease was observed in other mice that were orally fed with recPrP^Sc^, DBH or sPMCA substrate. All mice were sacrificed at 650 dpi.

Mouse brain homogenates were prepared and subjected to PK digestion. Immunoblot analysis revealed that the classic PK-resistant PrP^Sc^ was detected in the positive control mice that received i.c. prion inoculation, the #9 and #19 mice that were orally fed with recPrP^Sc^ and DBH, respectively, but not in other mice that were orally fed with recPrP^Sc^ or DBH (Figure 1C). Histopathological analyses revealed classical pathological changes in #9 and #19 mice, including spongiosis, astrogliosis, microgliosis and the accumulation of PK-resistant PrP (Figure 2).

### 2.2. Second Round Transmission

If #9 and #19 mice indeed developed prion disease, the disease should be able to serially transmit in mice. In addition, the lower infectivity of orally fed recPrP^Sc^ may create a subclinical state with prion infectivity generated in the brains of clinically normal mice [11,12,13]. Alternatively, the orally fed recPrP^Sc^ may be subject to truncation or alteration in conformation due to oral digestion, which may need further adaptation in vivo to become a fully infectious prion [14]. To test these possibilities, we selected three groups of mice that (1) developed classic characteristics of prion disease (#9 and #19); (2) developed clasping, but without other signs of prion disease (#10, #14 and #21); (3) appeared normal (#8 and #18). Wild-type C57BL/6 mice were intracerebrally inoculated with 1% brain homogenates prepared from these mice.

All mice that were inoculated with #9 or #19 mouse brain homogenates developed fatal neurodegeneration with a survival time of 191 ± 4.97 dpi and 187 ± 3.38 dpi, respectively (Figure 3A and Table 2). For mice that received the inoculation of other mouse brain homogenates, although clasping was observed in some of them, none of the mice developed typical signs of prion disease and all were sacrificed at 330 dpi (Figure 3A and Table 2). Biochemical analysis revealed that the PK-resistant PrP^Sc^ was only detected in mice that were inoculated with #9 or #19 mouse brain homogenates (Figure 3B). Consistent with this finding, spongiosis and PK-resistant PrP were only detected in mice that were inoculated with #9 or #19 brain homogenate, but not in mice that received other inocula (Figure 4).

## 3. Discussion

Since the first success in generating recPrP^Sc^ 10 years ago [4], recPrP^Sc^ has been extensively studied by multiple labs. However, its ability to cause prion disease via oral route had never been proven. Our study provided the first experimental evidence that oral feeding of recPrP^Sc^ is sufficient to cause prion disease in wild-type mice. Because orally ingested prions are exposed to the digestion process in the gastrointestinal tract that is known to degrade PrP^Sc^ [15], our results indicate that the in vitro generated recPrP^Sc^ is able to survive the digestion, maintain its infectious conformation, spread to the central nervous system and cause a fatal neurodegenerative disease in wild-type mice.

Although orally ingested recPrP^Sc^ successfully causes prion disease, the attack rate is low and only one out of 11 mice developed disease. The i.c. inoculation of the same batch of recPrP^Sc^, however, resulted in 100% attack rate and a rather synchronized survival time of 196.2 ± 3.59 days (Figure 1B and Table 1), indicating a higher infectivity. The reduced efficiency of orally fed recPrP^Sc^ is consistent with previous reports that compared to i.c. prion inoculation, the infectivity of oral dosing reduces by a factor ~10^5^ in mice [16] and ~10^9^ in hamster [17]. With gastric gavage, Sigurdson et al. showed that 6.4 LogLD_50_ infectious dose of RML prion caused prion disease in one of 11 mice [18]. Although we did not titrate the batch of recPrP^Sc^ used in this study, our previous study showed that a batch of similarly prepared recPrP^Sc^ contains ~10^4^ LD_50_ (by i.c. route)/µg of PrP [6], which indicates that the infectious dose of recPrP^Sc^ fed to a mouse in this study would be around 5 LogLD_50_ (by i.c. route). Even including the consideration of potential variations in each in vitro recPrP^Sc^ preparation, the dose used in our study would be similar or lower than that used by Sigurdson et al. [18]. Therefore, we concluded that the attack rate of orally fed recPrP^Sc^ is comparable to that of naturally occurring prions. Notably, the attack rate of orally fed DBH is only 1 out of 6 mice in our study, which is also significantly lower than that of i.c. inoculation (Figure 1B and Table 1).

In order to more stringently recapitulate the natural spread of prion via oral route, we chose to feed mice with recPrP^Sc^ instead of using gastric gavage, which would increase its exposure to the digestion and may also contribute to the low attack rate. In future studies, the infectivity of the inocula should be carefully titrated by i.c. inoculation. The dosage of orally fed recPrP^Sc^ can be increased and if necessary, the inocula can be concentrated by centrifugation. Together with increasing the size of the experimental group, these measures will allow us to accurately compare the rate of oral transmissibility of recPrP^Sc^ to that of DBH. In addition, a second round p.o. transmission may help to determine whether the disease can be serially transmitted via oral route, and whether the particular prion strain induced by orally fed recPrP^Sc^ has a preference for oral transmission. Despite these limitations of our study, the success of orally fed recPrP^Sc^ to cause prion disease in wild-type mice does support the idea that PrP^Sc^, the pathogenic conformer of PrP, is the cause for orally transmitted prion disease.

Several naturally occurring prion diseases, such as scrapie in sheep and goats and CWD in cervids, appear to be quite efficient in oral transmission [19,20,21], which may be attributed to the particular prion strains and animal species. Oral transmission of these natural prion diseases could also be enhanced by other factors, such as the binding of prion to soil particles [22], the presence of bacterial colitis [18], the alteration of intestinal M cells density regulated by RANKL [23] and lesions to the oral mucosa [24]. It will be interesting to determine whether any of these factors are able to enhance the oral transmission of recPrP^Sc^ in future studies.

Collectively, our study revealed that oral ingestion of in vitro generated recPrP^Sc^ is sufficient to cause prion disease in wild-type mice, which provides the first example that an in vitro generated pathogenic conformer of a recombinant protein is able to cause a fatal neurodegenerative disease via oral route. Together with previous findings, our results support that as the prion hypothesis postulated, the misfolded PrP conformer is responsible for the transmissibility of prion disease.

## 4. Materials and Methods

### 4.1. Mice

Wild-type C57BL/6 mice were purchased from the Shanghai Laboratory Animal Center (Shanghai, China) and 6-week-old female mice were used in this study. Mice were maintained under specific pathogen free (SPF) conditions. All mouse experiments were performed according to the Guidelines on the Humane Treatment of Lab Animals established in 2006 by the Ministry of Science and Technology of China [policy (2006)398] and approved by Center for Animal Experiment of Wuhan University and East China Normal University (m20160303).

### 4.2. Prion Exposure and Disease Monitoring

Preparation of inocula, i.c. inoculation, second-round i.c. transmission and the analyses of mouse brains were performed as previously described [5,25]. Oral exposure was performed according a previously reported protocol [23,26]. Briefly, a single food pellet was doused with 50 µL of sPMCA substrate, recPrP^Sc^ or 1% (*w*/*v*) DBH, and allowed to dry at room temperature. Mice were individually housed in bedding- and food-free cages during the oral feeding. A single food pellet was placed on the floor of the cage. Once the food pellet was completely ingested, the mouse was returned to its home cage. For i.c. inoculation, 30 µL of recPrP^Sc^ or 1% (*w*/*v*) DBH was injected into an anesthetized mouse. After oral feeding or i.c. inoculation, all mice were monitored three times a week. Once neurological signs were clearly identified, mice were monitored daily and euthanized at terminal disease stage. Efforts were made to minimize pain and suffering of animals.

### 4.3. Histopathological Analyses

Histopathological analyses were performed as previously described [25,27]. Hematoxylin & Eosin (H&E) staining was performed with Harris Hematoxylin (Sigma, St. Louis, MO, USA) and Eosin Y (Fisher, Waltham, MA, USA). For immunohistochemistry, 5-μm-thick paraffin sections were deparaffinized, hydrated, and stained with anti-glial fibrillary acidic protein (GFAP, #3670, 1:500, Cell Signaling Technology, Inc., Boston, MA, USA) or anti-ionized calcium-binding adaptor molecule 1 (Iba-1, 019-19741, 1:100, Wako Pure Chemical Industries, Ltd., Osaka, Japan) antibody. For PET blot, 5-μm-thick paraffin sections were collected onto 0.45-μm nitrocellulose membranes and incubated at 55 °C overnight. Membranes were deparaffinized and rinsed in isopropanol followed by hydration. The membranes were then subjected to 100 g/mL PK digestion in 10 mM Tris-HCl, pH 7.8, 100 mM NaCl and 0.1% Brij 35 for 16 h at 55 °C. After washing with tris-buffered saline with 0.1% tween 20 (TBST), the membrane was incubated in 4 M guanidine thiocyanate for 10 min and washed 3 times in TBST. The membrane was blocked by 2% non-fat milk in TBST (blocking solution) for 1 h, incubated with monoclonal SAF84 anti-PrP antibody (1:100 in blocking solution) for 90 min at room temperature followed by incubation with an alkaline phosphatase conjugated goat anti-mouse IgG antibody (1:500 in blocking solution) for 1 h at room temperature. The color was developed by using the 5-bromo-4-chloro-3-indolyl phosphate/nitro blue tetrazolium substrate and the images were captured with a digital camera.

### 4.4. Immunoblot Detection of PrP^Sc^

Brain homogenates (10%, *w*/*v*) were prepared and incubated with 25 µg/mL PK at 37 °C for 1 h. Digestion was terminated by adding 1 mM phenylmethysulfonyl fluoride. Samples were separated by electrophoresis through 12% Tris-glycine polyacrylamide gels and transferred to polyvinylidene difluoride membranes. PrP was detected with 3F10 anti-PrP monoclonal antibody (1:1500) [28].

## Figures and Tables

**Figure 1 pathogens-09-00653-f001:**
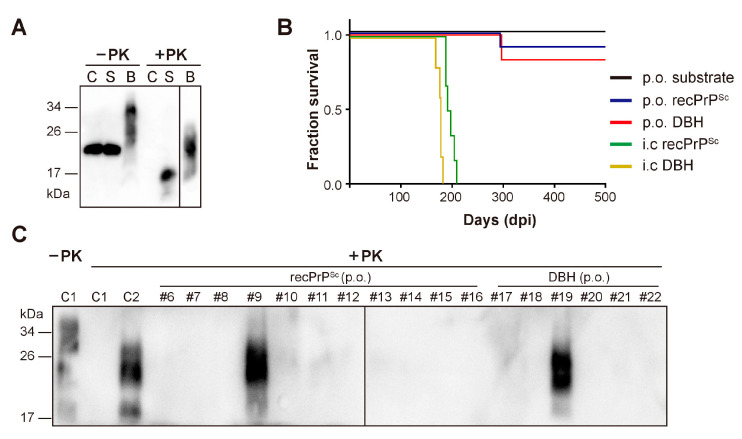
Oral (p.o.) or i.c. infection of wild-type C57BL/6 mice. (**A**) The presence of PK-resistant PrP in the inocula was detected by immunoblot analysis with 3F10 anti-PrP antibody. C, sPMCA substrate; S, recPrP^Sc^; B, prion diseased mouse brain homogenate. Samples were digested with 25 µg/mL PK at 37 °C for 30 min (for C and S) or 60 min (for B). (**B**) Survival curve of mice that received indicated inocula via i.c. or p.o. route. DBH, prion diseased mouse brain homogenate. (**C**) The presence of PK-resistant PrP in mouse brain homogenates was detected by immunoblot analysis with 3F10 anti-PrP antibody after 25 µg/mL PK digestion at 37 °C for 60 min. C1, a negative control mouse that was orally fed with sPMCA substrate; C2, a positive control mouse that received i.c inoculation of recPrP^Sc^. Mice that received oral feeding of recPrP^Sc^ (mouse #6–#16) or DBH (mouse #17–#22) were indicated.

**Figure 2 pathogens-09-00653-f002:**
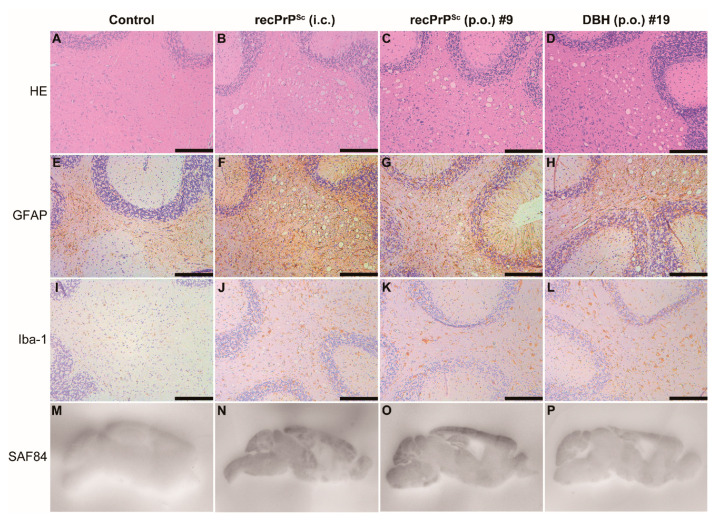
Pathological changes in mice that received first round prion inoculation. Histological and PET blot analyses of a negative control mouse that received oral feeding of sPMCA substrate (**A**,**E**,**I**,**M**), a positive control mouse that was i.c. inoculated with recPrP^Sc^ (**B**,**F**,**J**,**N**), the #9 mouse that received oral feeding of recPrP^Sc^ (**C**,**G**,**K**,**O**) and the #19 mouse that received oral feeding of DBH (**D**,**H**,**L**,**P**). Brain sections were stained by hematoxylin and eosin (HE) (**A**–**D**), immunohistochemistry with an anti-GFAP antibody (**E**–**H**) or anti-Iba-1 antibody (**I**–**L**), and PET blot with SAF84 anti-PrP antibody (**M**–**P**). Immunohistochemical stains were counterstained with hematoxylin. Scale bar represents 200 µm.

**Figure 3 pathogens-09-00653-f003:**
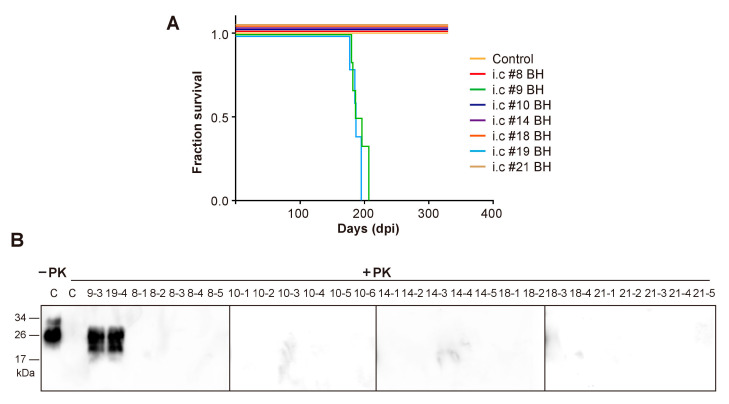
Second round transmission. (**A**) Survival curve of wild-type C57BL/6 mice that received i.c. inoculation of brain homogenates (BH) prepared from indicated mice. Control, mice that received i.c. inoculation of brain homogenate prepared from mice which were orally fed with sPMCA substrate in the first round. (**B**) PK-resistant PrP^Sc^ in mouse brain homogenates was detected by immunoblot analysis with 3F10 anti-PrP antibody after 25 µg/mL PK digestion at 37 °C for 60 min. C, a control mouse that was i.c. inoculated with brain homogenate prepared from a negative control mouse which was orally fed with sPMCA substrate in the first round. #9–3 and #19–4 are representative mice that were i.c. inoculated with brain homogenates prepared from #9 and #19 mice, respectively. For mouse numbering, the first letter represents the first round mouse number.

**Figure 4 pathogens-09-00653-f004:**
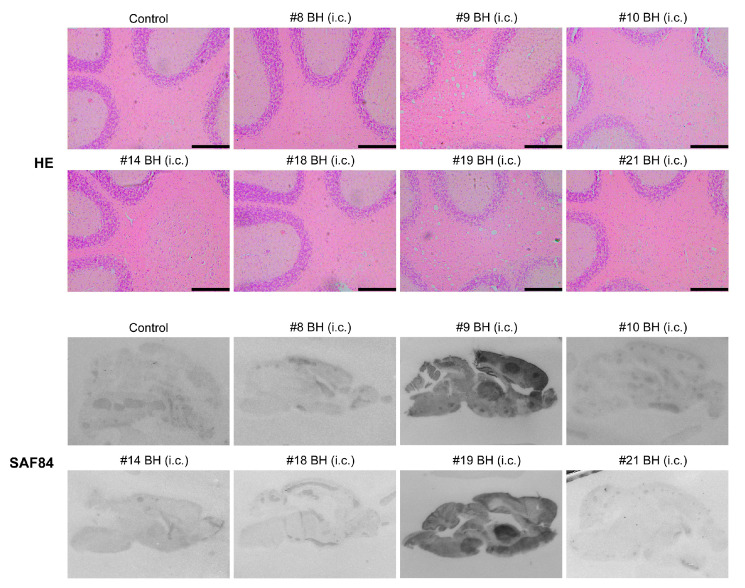
Pathological changes in mice that received second round transmission. As indicated, mouse brains were subjected to HE stain or PET blot analysis to determine the pathological changes. Control is a negative control mouse that was i.c. inoculated with brain homogenate prepared from a negative control mouse in the first round. Mice were indicated by the first round mouse number. BH, brain homogenates. Scale bar represents 200 µm.

**Table 1 pathogens-09-00653-t001:** First round transmission in wild-type C57BL/6 mice.

Inoculum	Route of Transmission	Diseased/Exposed Mice	Survival (Days) (Mean ± SEM)
sPMCA substrate	Oral feeding	0/5	>650
recPrP^Sc^	Oral feeding	1/11	294; 574 *; >650
DBH	Oral feeding	1/6	297; 586 *; >650
recPrP^Sc^	i.c inoculation	6/6	196.2 ± 3.59
DBH	i.c inoculation	5/5	176.4 ± 5.17

* Mice died of intercurrent diseases. DBH, prion diseased mouse brain homogenates.

**Table 2 pathogens-09-00653-t002:** Second round transmission in wild-type C57BL/6 mice.

Inoculum	Route of Transmission	Diseased/Inoculated Mice	Survival (Days) (Mean ± SEM)
Control	i.c.	0/5	>330
#8 mouse BH	i.c.	0/5	>330
#9 mouse BH	i.c.	6/6	191 ± 4.97
#10 mouse BH	i.c.	0/6	>330
#14 mouse BH	i.c.	0/5	>330
#18 mouse BH	i.c.	0/4	>330
#19 mouse BH	i.c.	5/5	187 ± 3.38
#21 mouse BH	i.c.	0/5	>330

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
