# Peer review of "Oral Ingestion of Synthetically Generated Recombinant Prion Is Sufficient to Cause Prion Disease in Wild-Type Mice"

_pathogens, 2020, doi:10.3390/pathogens9080653_

Round 1

Reviewer 1 Report

This is a small and uncomplicated study in prion transmission properties of prions produced in vitro. The authors do acknowledge previous successes in demonstrating the transmissibility of synthetic prions by the ip and ic routes of inoculation. The aim of this study was simply to investigate if the oral route of infection could be used to demonstrate the transmissibility of recombinant prions (recPrPSc). The justification for doing this study is acceptable because naturally occurring prion infections occur predominantly via the oral route. The positive rates of transmissions seen at first orally challenged passage in wild type mice were very low at 1/11 (recPrPSc synthetic prions) and 1/6 (BDH-positive control). However, the histopathological changes coupled with the enhanced rate of transmission at second passage confirmed the propagation of prions during the first passage. These low attack rates for both the recombinant and positive control isolates are consistent with the fact that the oral route of infection is known to be relatively inefficient in comparison to i.c and i.p routes. This finding makes a modest contribution to the literature, and there is no reason not to publish these results, subject to some revisions.

Main comments:

1) Figure 1 C: the sample in the first lane needs to be labelled and properly referenced in the legend, whether it is control PrPC or even a marker lane that has run poorly.

2) Figure 3 A: there are only 3 coloured lines on the survival curve but 8 colours in the key. If the authors have decided to show the curves of only 3 samples (control, #9 BH and #19 BH), then they need to show only these 3 samples in the key, and tell the reader in the legend that the remaining 5 are not shown. The more appropriate option is to plot the other 5 missing colours as well, to run parallel to the control, so that there are 8 coloured lines on the graph to match the 8-colour key. This a matter of data accuracy and they may want to look at Figure 1 B for a cue on how to maintain consistency in the reporting of their data.

Minor comments.

1) Line 63-64: “Brain homogenates prepared from a mouse succumbed to prion disease (DBH) was used as a positive control”. The acronym BDH is not properly defined at its first use. One can guess that BH stands for brain homogenate, and that D probably stands for “diseased”. But since it cannot be directly inferred by the reader from the sentence (for example PCR stands for Polymerase Chain Reaction and PMCA stands for Protein Misfolding Cyclic Amplification), the authors should consider making it clear that BDH is only a designation they have chosen for the positive control for simplicity, and not because BDH can be easily deduced directly from the sentence. The authors only need to write (designated BDH) at the first use. In the same sentence “mouse succumbed” should really be “mouse that succumbed”.

2) The standard of the writing is good, but this manuscript could still benefit from copy editing. The way that the word “received” has been used throughout the manuscript needs a second look in order to lift the standard of writing further and more importantly, show greater clarity of expression.

Two examples below:

1) Figure 1 legend (B) “Survival curve of mice received indicated inoculations via i.c. or p.o. route.” This sentence leaves one expecting something more to follow, but there is nothing else. The sentence is missing a single word (see below) that can fix the unintended double meaning.

2) Figure 2 title “Pathological changes in mice received first round prion inoculation”. This could be interpreted as ‘it is the pathological changes” that received first round prion inoculation, not the mice as intended. The inaccuracy in these sentences, and everywhere else in the manuscript where “received” has been used, can be corrected in 2 ways (1) insert “that or which” as in pathological changes in the mice that received…or (2) change received to receiving.

This style of using the word “received” throughout without prefacing with “that” or “which” makes the sentences awkward, to say the least (see lines 88, 91,92, 95, 97, 102, 103, 104, 120, 125, 127, 128, 129 -just too many to list all of them here). Clearly the authors are not aware of this systematic defect in sentence precision, but one hopes that a Copy Editor would act accordingly.

Other omissions/typos can be found at:

Line 112: may subject to (instead of ‘may be subject to’..)

Line 138: were subject to (instead of ‘were subjected to’, as used correctly in line 94)

Line 149: able to survival the digestion (instead of ‘able to survive’..)

Line 163: the attack rate orally fed recPrPSc is.. (instead of ’rate of’…as written correctly in line 164)

Line 172: which may attribute to the particular (instead of ‘may be attributed to’…)

Line 196: was place on the floor (instead of ‘was placed’…)

These are just what I did see without trying too hard. I would therefore advise a thorough revision of this manuscript either by the authors or a copy editor before publication, in order to enhance the reputation of the journal.

Author Response

Point-to-point response to reviewers’ comments

Reviewer 1

Main comments:

1) Figure 1 C: the sample in the first lane needs to be labelled and properly referenced in the legend, whether it is control PrPC or even a marker lane that has run poorly.

Response: We are sorry for the mistake. This is the undigested brain homogenate of the negative control mouse (C1). It is now properly labelled in the revised figure 1.  

2) Figure 3 A: there are only 3 coloured lines on the survival curve but 8 colours in the key. If the authors have decided to show the curves of only 3 samples (control, #9 BH and #19 BH), then they need to show only these 3 samples in the key, and tell the reader in the legend that the remaining 5 are not shown. The more appropriate option is to plot the other 5 missing colours as well, to run parallel to the control, so that there are 8 coloured lines on the graph to match the 8-colour key. This a matter of data accuracy and they may want to look at Figure 1 B for a cue on how to maintain consistency in the reporting of their data.

Response: We thank the reviewer for an excellent suggestion. All curves are properly shown in parallel to the control in the revised figure 3. In addition, we also revised the survival curve in figure 1B to keep the consistency.

Minor comments.

1) Line 63-64: “Brain homogenates prepared from a mouse succumbed to prion disease (DBH) was used as a positive control”. The acronym BDH is not properly defined at its first use. One can guess that BH stands for brain homogenate, and that D probably stands for “diseased”. But since it cannot be directly inferred by the reader from the sentence (for example PCR stands for Polymerase Chain Reaction and PMCA stands for Protein Misfolding Cyclic Amplification), the authors should consider making it clear that BDH is only a designation they have chosen for the positive control for simplicity, and not because BDH can be easily deduced directly from the sentence. The authors only need to write (designated BDH) at the first use. In the same sentence “mouse succumbed” should really be “mouse that succumbed”.

Response: In lines 64-65 of the revised manuscript, we changed the sentence to “Brain homogenates prepared from a mouse that succumbed to prion disease (DBH, representing prion diseased brain homogenates) was used as a positive control.”

2) The standard of the writing is good, but this manuscript could still benefit from copy editing. The way that the word “received” has been used throughout the manuscript needs a second look in order to lift the standard of writing further and more importantly, show greater clarity of expression.

Two examples below:

1) Figure 1 legend (B) “Survival curve of mice received indicated inoculations via i.c. or p.o. route.” This sentence leaves one expecting something more to follow, but there is nothing else. The sentence is missing a single word (see below) that can fix the unintended double meaning.

2) Figure 2 title “Pathological changes in mice received first round prion inoculation”. This could be interpreted as ‘it is the pathological changes” that received first round prion inoculation, not the mice as intended. The inaccuracy in these sentences, and everywhere else in the manuscript where “received” has been used, can be corrected in 2 ways (1) insert “that or which” as in pathological changes in the mice that received…or (2) change received to receiving.

This style of using the word “received” throughout without prefacing with “that” or “which” makes the sentences awkward, to say the least (see lines 88, 91,92, 95, 97, 102, 103, 104, 120, 125, 127, 128, 129 -just too many to list all of them here). Clearly the authors are not aware of this systematic defect in sentence precision, but one hopes that a Copy Editor would act accordingly.

Response: We greatly appreciate the reviewer’s comment and fixed the problem as suggested.

Other omissions/typos can be found at:

Line 112: may subject to (instead of ‘may be subject to’..)

Line 138: were subject to (instead of ‘were subjected to’, as used correctly in line 94)

Line 149: able to survival the digestion (instead of ‘able to survive’..)

Line 163: the attack rate orally fed recPrPSc is.. (instead of ’rate of’…as written correctly in line 164)

Line 172: which may attribute to the particular (instead of ‘may be attributed to’…)

Line 196: was place on the floor (instead of ‘was placed’…)

These are just what I did see without trying too hard. I would therefore advise a thorough revision of this manuscript either by the authors or a copy editor before publication, in order to enhance the reputation of the journal.

Response: Again, we thank the reviewer for his careful review and for pointing out these mistakes. We fixed these mistakes and carefully went through the manuscript multiple times to check other writing mistakes.

Reviewer 2 Report

In this study, the authors demonstrate the infectivity of recPrPSc (generated by PMCA) in wildtype mice. This is observed in all cases after intracerebral inoculation, but only in 1 out of 11 mice that received oral administration. I do not know if this modest effect was the result expected by the authors. But since I do believe that the observed effect size should not be the main determinant to base the revision of a manuscript on, I will proceed with my comments:

1 – There are two facts that bring me to the suspicion that maybe the results of oral administration observed in this work are somehow in a “grey” zone. I mean, it seems reasonable to ascertain that prion oral feed induce prion disease, but there is still room to observe a stronger effect. On one side, the numbers are clear: 1 out of 11 mouse developed disease. On the other side, it is mentioned that 3 p.o. mice (#10, #14 and #21) developed clasping, which seems to be relevant enough for the authors (and for me) to use these animals’ material for the second round of infection .

Altogether, it may make sense to increase the dosage and/or the time exposure (I know 650 dpi is already at late mouse lifetime, but inoculation could have been performed earlier). Why was this not considered?

Maybe adding a little discussion about this possibility would increase the relevance of the only 1/11 effect that is presented.

Just a short note: from human prion diseases, we know that there is a long pre-clinical phase in which the pathological conversion of the PrP might have already started in the brain, before the onset of symptoms. That’s why I think that this study might have ended too early (little amounts of PrPSc might not be visible with histopathology or WB).

2- Were the neurological signs displayed by mice fed with recPrPSc and with DBH the same?

Histopathology images indicate similar brain affectation, but what about the phenotype developed during life time? Given the low number of animals, a table with a description of clinical symptomatology and time of onset would be informative to observe potential differences due to different infective material.

3- Why in the first round experiment, mice were sacrificed at 650 dpi and in the second round experiment at 330 dpi? Is there any justification?

4- Why was oral administration not considered in the second round experiment? Looking only at both survival plots, it seems that route of infection is what determines the survival time (~300 dpi with p.o., and ~180-190 with i.c.). And this seems to be independent from the infective material (DBH or recPrPSc). Caution should be taken because the number of observations per conditions is very small. But having used p.o. in the second round of infection would have allowed testing this hypothesis. I think a short discussion on future research lines, or justifying why this was not done would make the discussion more complete.

5- Indeed, the discussion is the weakest part in my opinion. I have some suggestions:

5.1. Adding a limitation section (for example, the low number of animals, the lack of quantification of PrPSc in the inocula,… )

5.2. Comparing the attack rate of orally fed recPrPSc and DBH, and stating it is a little bit higher, but in a similar range is rather speculative. With the low number of animals used and the results obtained 1/6 vs. 1/11, it is just possible to say that these rate were lower than those of i.c.. But the low number of animals makes impossible to extract any other conclusions statistically supported by data (for instance, attack rate could either be 20% or 1%, impossible to guess it).

5.3. I miss a discussion on the implications of these findings. Besides the “strong experimental evidence to support prion hypothesis”, which is already demonstrated by many others, I think there are implications for biosafety. If PMCA products are infective just with oral administration, what about the products of RT-QuIC for instance? (which in many diagnostic labs are treated as P1 biosafety level material). Some discussion about these biosafety issues could be of interest to the readers.

6- Formal aspects:

6.1- In fig 3A, there are many overlapping observations, so basically only 3 colours remain visible. Please present the plot in a way that results more clearly, or if necessary, group all mice that did not die in one single line colour/format.

6.2- In fig 3B, “C2” and “C3” nomenclature creates misunderstanding, because these lanes are not control lanes, (if I’m not wrong) they just refer to the positive results obtained with mice #9 #19 homogenates. I think it is better to follow a uniform nomenclature and refer to these lanes the same way as the other lanes (the first letter representing the 1st round mouse number)

Author Response

Point-to-point response to reviewers’ comments

Reviewer 2

1 – There are two facts that bring me to the suspicion that maybe the results of oral administration observed in this work are somehow in a “grey” zone. I mean, it seems reasonable to ascertain that prion oral feed induce prion disease, but there is still room to observe a stronger effect. On one side, the numbers are clear: 1 out of 11 mouse developed disease. On the other side, it is mentioned that 3 p.o. mice (#10, #14 and #21) developed clasping, which seems to be relevant enough for the authors (and for me) to use these animals’ material for the second round of infection .

Altogether, it may make sense to increase the dosage and/or the time exposure (I know 650 dpi is already at late mouse lifetime, but inoculation could have been performed earlier). Why was this not considered?

Maybe adding a little discussion about this possibility would increase the relevance of the only 1/11 effect that is presented.

Response: These are great suggestions. In the discussion of the revised manuscript, we discussed these possibilities. 

Just a short note: from human prion diseases, we know that there is a long pre-clinical phase in which the pathological conversion of the PrP might have already started in the brain, before the onset of symptoms. That’s why I think that this study might have ended too early (little amounts of PrPSc might not be visible with histopathology or WB).

Response: We agree with the reviewer and that is exactly the reason for us to perform the second round transmission.  

2- Were the neurological signs displayed by mice fed with recPrPSc and with DBH the same?

Histopathology images indicate similar brain affectation, but what about the phenotype developed during life time? Given the low number of animals, a table with a description of clinical symptomatology and time of onset would be informative to observe potential differences due to different infective material.

Response: The clinical manifestations are essentially the same, except that urinary incontinence was only observed in the #19 mouse fed with DBH. Because there is only one mouse in each group developed disease, it is difficult for us to draw any conclusion. We added this sentence in the revised manuscript (lines 80-81), “Both mice developed unsteady gait, tail plasticity, kyphosis and weight loss, but urinary incontinence was only observed in #19 mouse.”

3- Why in the first round experiment, mice were sacrificed at 650 dpi and in the second round experiment at 330 dpi? Is there any justification?

Response: The main purpose of the first round is to determine the oral transmissibility, which is usually low, resulting in longer incubation time and reduced attack rate. Longer time is to ensure that we do not miss any mice that may develop disease. The main purpose for the second round transmission is to determine whether there is prion infectivity in these mouse brains. Since the survival time for i.c. inoculated prion is generally around 180-190 dpi, 330 dpi is generally considered to be sufficient to detect prion infectivity.   

4- Why was oral administration not considered in the second round experiment? Looking only at both survival plots, it seems that route of infection is what determines the survival time (~300 dpi with p.o., and ~180-190 with i.c.). And this seems to be independent from the infective material (DBH or recPrPSc). Caution should be taken because the number of observations per conditions is very small. But having used p.o. in the second round of infection would have allowed testing this hypothesis. I think a short discussion on future research lines, or justifying why this was not done would make the discussion more complete.

Response: Because of the purpose of second round transmission (as stated above) and the consideration of time and financial cost, we only performed i.c. inoculation in the second round and terminated the study at 330 dpi. As the reviewer suggested, we include the following sentence in the discussion. “In addition, a second round p.o. transmission may help to determine whether the disease can be serially transmitted via oral route, and whether the particular prion strain induced by orally fed recPrPSc has a preference for oral transmission.” (Lines 203-205 of the revised manuscript)

5- Indeed, the discussion is the weakest part in my opinion. I have some suggestions:

5.1. Adding a limitation section (for example, the low number of animals, the lack of quantification of PrPSc in the inocula,… )

Response: We thank the reviewer for the excellent suggestion. A paragraph is added in lines 199-205, “In future studies, the infectivity of the inocula should be carefully titrated by i.c. inoculation. The dosage of orally fed recPrPSc can be increased and if necessary, the inocula can be concentrated by centrifugation. Together with increasing the size of the experimental group, these measures will allow us to accurately compare the rate of oral transmissibility of recPrPSc to that of DBH. In addition, a second round p.o. transmission may help to determine whether the disease can be serially transmitted via oral route, and whether the particular prion strain induced by orally fed recPrPSc has a preference for oral transmission.”

5.2. Comparing the attack rate of orally fed recPrPSc and DBH, and stating it is a little bit higher, but in a similar range is rather speculative. With the low number of animals used and the results obtained 1/6 vs. 1/11, it is just possible to say that these rate were lower than those of i.c.. But the low number of animals makes impossible to extract any other conclusions statistically supported by data (for instance, attack rate could either be 20% or 1%, impossible to guess it).

Response: We agree with the reviewer. The paragraph is now changed to “Notably, the attack rate of orally fed DBH is only 1 out of 6 mice in our study, which is also significantly lower than that of i.c. inoculation (Fig 1B and table 1).” (lines 194-196 of the revised manuscript).

5.3. I miss a discussion on the implications of these findings. Besides the “strong experimental evidence to support prion hypothesis”, which is already demonstrated by many others, I think there are implications for biosafety. If PMCA products are infective just with oral administration, what about the products of RT-QuIC for instance? (which in many diagnostic labs are treated as P1 biosafety level material). Some discussion about these biosafety issues could be of interest to the readers.

Response: Biosafety is a concern, but the product of RT-QuIC has never been proved to be infectious in animals. In addition, the newer RT-QuIC uses hamster PrP90-231, which further improves the safety. That been said, we do understand reviewer’s concern. We added a sentence in the discussion to emphasize the point that in vitro conversion of normal recombinant protein can become a disease-causing agent. “…which provides the first example that an in vitro generated pathogenic conformer of a recombinant protein is able to cause a fatal neurodegenerative disease via oral route.” (Lines  236-238 of the revised manuscript)

6- Formal aspects:

6.1- In fig 3A, there are many overlapping observations, so basically only 3 colours remain visible. Please present the plot in a way that results more clearly, or if necessary, group all mice that did not die in one single line colour/format.

Response: See response to reviewer 1’s main comment #2.

6.2- In fig 3B, “C2” and “C3” nomenclature creates misunderstanding, because these lanes are not control lanes, (if I’m not wrong) they just refer to the positive results obtained with mice #9 #19 homogenates. I think it is better to follow a uniform nomenclature and refer to these lanes the same way as the other lanes (the first letter representing the 1st round mouse number)

Response: We agree. They are now changed to 9-3 and 19-4.